# Contribution of the Locus of Heat Resistance to Growth and Survival of *Escherichia coli* at Alkaline pH and at Alkaline pH in the Presence of Chlorine

**DOI:** 10.3390/microorganisms9040701

**Published:** 2021-03-28

**Authors:** Tongbo Zhu, Zhiying Wang, Lynn M. McMullen, Tracy Raivio, David J. Simpson, Michael G. Gänzle

**Affiliations:** 1Department of Agricultural, Food and Nutritional Science, 4-10 Ag/For Centre, University of Alberta, Edmonton, AB T6G 2P5, Canada; tongbo@ualberta.ca (T.Z.); zhiying2@ualberta.ca (Z.W.); lmcmulle@ualberta.ca (L.M.M.); djsimpso@ualberta.ca (D.J.S.); 2Department of Biological Science, University of Alberta, Edmonton, AB T6G 2E9, Canada; traivio@ualberta.ca

**Keywords:** *Escherichia coli*, locus of heat resistance, Cpx two-component regulatory system, alkaline pH response

## Abstract

The locus of heat resistance (LHR) confers resistance to extreme heat, chlorine and oxidative stress in *Escherichia coli*. This study aimed to determine the function of the LHR in maintaining bacterial cell envelope homeostasis, the regulation of the genes comprising the LHR and the contribution of the LHR to alkaline pH response. The presence of the LHR did not affect the activity of the Cpx two-component regulatory system in *E. coli*, which was measured to quantify cell envelope stress. The LHR did not alter *E. coli* MG1655 growth rate in the range of pH 6.9 to 9.2. However, RT-qPCR results indicated that the expression of the LHR was elevated at pH 8.0 when CpxR was absent. The LHR did not improve survival of *E. coli* MG1655 at extreme alkaline pH (pH = 11.0 to 11.2) but improved survival at pH 11.0 in the presence of chlorine. Therefore, we conclude that the LHR confers resistance to extreme alkaline pH in the presence of oxidizing agents. Resistance to alkaline pH is regulated by an endogenous mechanism, including the Cpx envelope stress response, whereas the LHR confers resistance to extreme alkaline pH only in the presence of additional stress such as chlorine.

## 1. Introduction

*Escherichia coli* are commensals in the intestine of humans and animals, but the species also includes pathogenic strains that cause infections of the gastrointestinal and urinary tracts [1,2]. Food contamination by pathogenic *E. coli* may occur at any step of the farm-to-fork continuum and is a significant contributor to foodborne disease [3,4]. In food processing plants, alkaline-chlorinated sanitizers are used to sanitize food contact surfaces [5]. The chlorine and alkaline treatments function through synergistic mechanisms. Hypochlorous acid (HOCl), the active component of chlorine, oxidizes cellular components and permeabilizes the cytoplasmic membrane [6,7,8]. Extreme alkaline pH also denatures proteins and results in membrane permeabilization [9]. Mechanisms that allow strains of *E. coli* to resist chlorine include the locus of heat resistance (LHR) [10,11].

The LHR is a genomic island flanked by mobile genetic elements, which transfers among diverse species of *Enterobacteriaceae*, including opportunistic human pathogens in the genera *Cronobacter*, *Klebsiella* and *Enterobacter* [12,13]. The most frequent LHR variant in *E. coli*, including *E. coli* AW1.7, is the 15 kb LHR1, but other sequence variants with insertions or deletions, including the 19 kb LHR2, were also described [12,13]. Collectively, genes encoded by the LHR confer resistance to heat, chlorine and oxidative stress by reducing protein aggregation, protein oxidation and the oxidation of membrane lipids [10,14,15,16]. Proteomic analysis demonstrated that 11 of the 16 putative open reading frames of the LHR1 are expressed; highly expressed proteins include sHsp20, ClpK_GI_, sHsp_GI_, YfdX1_GI_ and YfdX2 [16]. The small heat shock protein sHsps and disaggregase ClpK prevent the aggregation of misfolded proteins or disaggregate and refold denatured proteins [17,18]. YfdX has a predicted signal peptide in the N-terminus, is likely localized in the periplasmic space, demonstrating chaperone-like activity [19], and is induced by the EvgA response regulator [20]. The LHR also encodes for KefB_GI_, a Na^+^/H^+^ antiporter that may function to maintain a polarized membrane in alkaline conditions [14] and thus may contribute to alkaline resistance, i.e., growth at alkaline pH, or alkaline tolerance, i.e., improved survival after lethal challenge with alkaline [21,22]; however, the role of KefB_GI_ in alkaline tolerance or alkaline resistance has not been verified experimentally.

Although close homologs of several highly expressed LHR-encoded proteins have been shown or implied to function on the bacterial cell envelope, the ability of the LHR to mitigate envelope stress remains to be investigated [16,19,23]. In *E. coli*, homeostasis of the cell envelope in response to perturbations is provided by the envelope stress responses. Cpx, a two-component regulatory system consisting of CpxA, a membrane-associated sensor kinase and the response regulator CpxR, is a widely conserved regulator of the envelope stress response in Gram-negative bacteria [24,25]. The most commonly accepted model suggests that the Cpx system detects misfolded proteins in the periplasm and maintains the integrity of the cytoplasmic membrane [26,27]. Therefore, Cpx is a suitable reporter system to monitor putative protective effects of LHR-encoded proteins on the cell envelope. In addition, the functional overlap between the LHR and Cpx response also suggests the possibility of cross-talk between these two mechanisms of stress resistance. This study therefore aimed to determine whether the LHR impacts the expression of the Cpx regulon, whether CpxR regulates the expression of the genes encoded by the LHR, and to compare the role of CpxR and the LHR on the growth and survival of *E. coli* at alkaline pH.

## 2. Materials and Methods

### 2.1. Bacterial Strains, Plasmids and Growth Conditions

Strains and plasmids used in this study are listed in Table 1. Unless otherwise stated, *Escherichia coli* strains were grown at 37 °C on Luria–Bertani (LB) plates or in LB broth with antibiotics added when necessary.

### 2.2. Construction of Derivatives of E. coli MG1655

The strain *E. coli* MG1655 used in this study is an *E. coli* K-12 derivative. The strain MG1655 *lacZ*::LHR with a chromosomal integration of the LHR was constructed by the Scarless Cas9-Assisted Recombineering system [29]. pLHR was digested by the DraI enzyme (Thermo Fisher, Mississauga, ON, Canada) to linearize the pLHR plasmid, which contains the LHR fragment flanked by parts of the *lacZ* gene [12]. The λ-Red was preinduced in an MG1655 strain containing both pKDsg-lacZ and pCas9cr4 followed by electroporation of the LHR fragments. After recovery, the culture was grown overnight with chloramphenicol (34 mg/L), spectinomycin (50 mg/L) and anhydrotetracycline (100 µg/L). The overnight culture was treated at 60 °C for 5 min and plated on LB agar containing IPTG (isopropyl β-d-1-thiogalactopyranoside, 0.2 mM) and X-gal (40 g/L). White colonies were screened with LHR-16-F/lacZ-upstream, LHR-2-R/lacZ-downstream and *yfdX1*-check-F/R primers (Table 2) to identify the target mutant followed by plasmid loss.

MG1655 *lacZ*::LHR Δ*kefB_GI_* was constructed using the λ-Red system [31]. The donor DNA was PCR generated with a chloramphenicol cassette flanked by sequences up- and downstream of *kefB_GI_*. pKD46 was transformed into MG1655 *lacZ*::LHR, and λ-Red was induced to facilitate the genomic integration of donor DNA. The mutant was picked from plates containing chloramphenicol (25 mg/L) and checked with primers listed in Table 2. pCP20 was used to remove the chloramphenicol cassette [31].

Mutant constructions of Δ*cpxR* and Δ*evgA* were done by P1 transduction based on the previous description [32]. Briefly, the P1 *vir* phage was first grown on the donor KEIO collection single-gene knockout strain, and the resulting P1 lysate was used to infect the recipient strains [33]. The potential mutants were selected on LB plates containing low-concentration kanamycin (25 mg/L) followed by purification with a high concentration (50 mg/L). Mutations were confirmed by PCR using the primers listed in Table 2.

### 2.3. Phylogenetic Tree of the CpxR Response Regulator

To construct a phylogenetic tree, the amino acid sequence of CpxR from *E. coli* MG1655 was used as the query to blast against the National Center for Biotechnology Information (NCBI) non-redundant reference proteins database in 15 May 2020. A coverage of 70% and identity of 60% were used as the cut-off values. The tree was generated with one sequence from each species of the family *Enterobacteriaceae* with the sequence from *Vibrio cholerae* as outgroup. The sequences were aligned by MUSCLE (https://www.ebi.ac.uk/Tools/msa/muscle/; accessed on 4 June 2020). The tree was constructed using maximum likelihood by MEGAX [36] (v10.1.8) and viewed by iTol (https://itol.embl.de; accessed on 5 June 2020).

### 2.4. Determination of the Cpx Pathway by Bioluminescence Assay

The Cpx response was determined with a luminescent reporter assay [37]. Single colonies of MG1655 and MG1655 *lacZ*::LHR carrying the low-copy-number plasmids pJW15 or pJW25 were inoculated overnight in LB broth and subcultured 1:100 into 5 mL of LB with 25 mg/L kanamycin and incubated with agitation at 200 rpm. Different from all other assays, which used an incubation temperature of 37 °C, all cultures used for the lux assay were incubated at 30 °C because the optimal temperature for luciferase is 28 °C. The Cpx activities were induced in different conditions. Cultures growing exponentially at 30 °C were harvested at an OD_600nm_ of 0.4; stationary-phase cultures were harvested at OD_600nm_ of 1.0. To determine the influence of aeration on Cpx activity, half the volume of each subculture was cultivated in aerobic conditions; the other aliquot was incubated in a 2 mL microcentrifuge tube that was closed and additionally sealed with parafilm to reduce diffusion of oxygen. The subcultures were incubated together and incubated at 30 °C 200 rpm for 18 h before measurements. To determine the impact of alkaline pH, cells from 2 mL culture were harvested and resuspended in 2 mL of LB with pH adjusted to 7.0, 8.0, 8.5 and 9.0 by adding HCl or NaOH. The bacteria were induced for 1 h in alkaline conditions before readings were taken. For other experiments, exponentially growing bacteria were induced for 2 h with the following inducing reagents (final concentration): 1 mM ZnSO_4_, 1 mM CuSO_4_ and 6 mM 2-phenylethanol (Sigma-Aldrich, Oakville, ON, Canada). Cultures without an inducing agent served as a reference. Luminescence and optical density at 600 nm of a 200 μL aliquot were measured in 96-well clear-bottom white plates (Thermo-Fisher) by plate readers (luminescence: Victor X4, Perkin Elmer, Waltham, MS, USA; OD_600_: Varioskan Flash, Thermo Fisher). The normalized luminescence value was calculated by standardizing the luminescence intensity to the OD_600_. The values obtained from the strains with pJW15 were used as a blank and subtracted from the readings acquired from the same strains containing pJW25. The fold change was calculated by dividing values for the induced conditions by the values for the uninduced condition of the same strain. Bioluminescence assays were performed in triplicate independent experiments that were analyzed in duplicate.

### 2.5. Measurement of LHR Gene Expression by RT-qPCR

The expression of genes encoded by the LHR was quantified by RT-qPCR essentially as previously described [10]. Overnight cultures were subcultured 1:100 in LB buffered to a pH of 7.1 or 8.2 by the addition of Tris and phosphate (50 mM each) and were grown to an OD_600nm_ of 0.5 at 37 °C with 200 rpm agitation. The cells were harvested, and RNA isolation was done using RNAprotect bacteria reagent and the RNAeasy mini kit (Qiagen, Hilden, Germany). The gDNA was removed by RQ1 RNase-Free DNase (Promega, Madison, WI, USA) and the reverse transcription of the RNA to cDNA was done with the QuantiTect^®^ reverse transcription kit (Qiagen). QuantiTect SYBR^®^ Green PCR Kits (Qiagen) and the 7500 fast real-time PCR system (Applied Biosystems, Waltham, MS, USA) were used to measure the LHR expression with the primers targeting *orf1*, *yfdX1_GI_* and *kefB_GI_* (Table 2). The glyceraldehyde-3-phosphate dehydrogenase A gene (*gapA*) was used as the house-keeping gene; DNase-digested RNA and water served as negative controls. The log_2_-normalized relative gene expression level was calculated by the ΔΔC_T_ method [10] by calculating gene expression by MG1655 *lacZ*::LHR Δ*cpxR* or MG1655 *lacZ*::LHR Δ*evgA* relative to the expression of the same gene by MG1655 *lacZ*::LHR grown at the same pH. Data shown are from three independent experiments that were analyzed in duplicate.

### 2.6. Determination of the Growth Rates

Tris-phosphate (50 mM each)-buffered LB was prepared at two different pH levels each at two different osmolarity levels and filter sterilized. Unless otherwise noted, LB media contained 10 g/L NaCl. The low-salt media was made by adding 10 g of tryptone, 5 g of yeast extract, 4.1 g of NaCl, 6.1 g of trizma base and 7.1 g of Na_2_HPO_4_ (Sigma-Aldrich) to 1 L of distilled water with a final pH of 6.5 and 10.0. To obtain the working media, LB broth at each pH and osmolarity were mixed in different ratios in clear-bottom 96-well plates (Corning, USA) to achieve a pH gradient. The resulting pH in media containing the high pH and low-pH media in different ratios was measured with a glass electrode. The bacteria were grown for 24 h in 5 mL of LB broth and subcultured 1:1000 in a final volume of 155 μL. OD_600nm_ was measured every 30 min for 16 h by a microtiter plate reader at 37 °C with a rotation diameter of 6 mm/s. Growth rates were calculated by fitting experimental data to the logistic growth curve [22] in SigmaPlot (version 12.5, Systat Software Inc., San Jose, CA, USA) with the data from three biological replicates.

### 2.7. Determination of the Tolerance to Extreme Alkaline pH with or without Chlorine Treatment

To determine the bacterial tolerance to extreme alkaline pH, filter-sterilized carbonate-bicarbonate (50 mM)-buffered LB was prepared by adding 0.525 g of sodium bicarbonate and 4.637 g of sodium carbonate (anhydrous) into 1 L of LB broth with the final pH of 11.2. Cultures were grown overnight in LB at 37 °C at 200 rpm agitation; cells from 500 μL of culture were pelleted by centrifugation and resuspended in 500 μL of carbonate-bicarbonate-buffered LB for 5 min. Sterile water (4.5 mL of 18 MΏ water) was added to the mixture and the after-treatment pH was measured. Alkaline pH tolerance with the addition of chlorine was determined by resuspending the bacterial pellet the same volume of filter-sterilized pH 11 LB including 10 mM NaClO (5% *w/w*, Sigma-Aldrich). After 5 min, cells were harvested by centrifugation and resuspended in LB broth. The viable cell count was determined by serial dilution and surface plating using LB broth and agar plate for both treated and untreated samples. The values were log-transformed and the reduction of cell count was calculated as log(N_0_/N). Data are from three independent experiments performed with technical repeats.

### 2.8. Statistical Analysis

Statistical analyses were performed with RStudio (version 1.2.1335, R Core Team, Vienna, Austria). Technical repeats were averaged without carry-over of the error term prior to statistical analysis. Differences among Cpx activities and gene expression were determined by a one-sample, two-tailed Student’s *t*-test (*p* < 0.05). Differences among cell count reductions were determined by one-way ANOVA followed by Tukey’s HSD (*p* < 0.05).

## 3. Results

### 3.1. Coexistence of the CpxR Response Regulator and the LHR

In the Enterobacterales, the LHR has to date only been identified in the family *Enterobacteriaceae* and the genus *Yersinia* [12]. To provide an initial assessment of cross-talk between CpxR and LHR, we determined whether genomes of those species that encode for the LHR also encode for CpxR. The CpxR response regulator is present in all species possessing the LHR, except *Citrobacter braakii*, indicating that interaction between the Cpx pathway and LHR is feasible (Figure 1).

### 3.2. The Presence of the LHR Does Not Alter Cpx Pathway Activity

To determine whether LHR plays a role in maintaining protein homeostasis in the bacterial periplasm or inner membrane integrity, we assessed the Cpx pathway activity in *E. coli* MG1655 and its derivative carrying the LHR, MG1655 *lacZ*::LHR, under various stress conditions. The Cpx response is activated by the stationary phase, aerobic conditions, elevated pH and the presence of zinc, copper and ethanol [27,38,39,40,41,42,43]. As a member of the Cpx regulon, *cpxP* is one of the most upregulated genes when Cpx is activated [40]. Therefore, pJW15, the promoterless vector control carrying *lux* operon and its derivative pJW25 with *cpxP’-lux* reporter were used to monitor Cpx activity. The Cpx pathway was highly upregulated after induction relative to the reference conditions in both *E. coli* MG1655 and *E. coli* MG1655 *lacZ*::LHR (Figure 2); however, this induction was not altered (*p* > 0.05) by the presence of the LHR (Figure 2).

### 3.3. LHR Transcriptional Level Is Affected by CpxR but Not by the EvgA Response Regulator at Alkaline pH

To determine whether the expression of the LHR is directly or indirectly dependent on the EvgA or CpxR response regulators, the mRNA of three fragments of the LHR, *orf1*, *yfdX1_GI_* and *kefB_GI_* [14], were quantified by RT-qPCR in *E. coli* MG1655 *lacZ*::LHR, MG1655 *lacZ*::LHR Δ*cpxR* and MG1655 *lacZ*::LHR Δ*evgA*. At pH 7.1, the difference in expression of the LHR among the strains was smaller than the experimental error that is generally observed for mRNA quantification with the ΔΔC_T_ method (Figure 3A). At a pH of 8.0, the lack of functional EvgA did not affect LHR transcription; however, deletion of *cpxR* increased the expression of *orf1*, *yfdX1_GI_* and *kefB_GI_* (Figure 3B). Results indicate that alkaline conditions activate LHR expression in the absence of the CpxR response regulator.

### 3.4. Cpx but Not the LHR Is Necessary for Growth in Alkaline pH

The Cpx response has been implicated in elevated pH adaptation, and the disruption of *cpxR* in *E. coli* impaired growth under alkaline conditions [40]. KefB is a glutathione-regulated potassium/proton antiporter-protecting bacteria from electrophile toxicity [44]. Therefore, to determine the role of the LHR in resistance to alkaline pH and the contribution of *kefB_GI_*, we assessed the growth rates of *E. coli* MG1655, MG1655 *lacZ*::LHR, MG1655 ΔcpxR, MG1655 *lacZ*::LHR Δ*cpxR* and MG1655 *lacZ*::LHR Δ*kefB_GI_* in buffered LB at pH 6.9 to 9.2. The growth rates of *E. coli* MG1655, MG1655 *lacZ*::LHR and MG1655 *lacZ*::LHR Δ*kefB_GI_* at neutral or alkaline were not different (Figure 4). The growth rates of Δ*cpxR* LHR-negative and -positive strains were comparable to the strains with a functional CpxR at pH values below 8.0. At pH 8.0 and above, deletion of *cpxR* reduced growth, and the maximum pH value of growth was reduced from 9.2 to 8.2, irrespective of the presence of the LHR (Figure 4). Comparable results were obtained in media containing 0.4 or 1% NaCl (Figure 4). Therefore, a functional Cpx pathway but not the LHR is required for bacterial growth at mild alkaline pH.

### 3.5. LHR Improves Bacterial Survival to Extreme Alkaline pH in the Presence of Chlorine

The LHR protects *E. coli* against lethal challenges with chlorine [10]; therefore, the role of CpxR and the LHR was explored by determination of the survival of *E. coli* MG1655, MG1655 *lacZ*::LHR, MG1655 Δ*cpxR*, MG1655 *lacZ*::LHR Δ*cpxR*, MG1655 *lacZ*::LHR Δ*kefB_GI_* and MG1655 *lacZ*::LHR Δ*cpxR* Δ*kefB_GI_* under extreme alkaline conditions in the absence or presence of chlorine. All strains survived challenge at pH 11.0 for 5 min with a reduction of viable cell counts of less than 1 log(cfu/mL); challenge at pH 11.3 reduced cell counts of all strains to levels below the detection limit (Appendix A). When treated with carbonate-bicarbonate-buffered LB at pH 11.2, the viable cell counts of all strains were reduced by about 1–2 log(cfu/mL) (Figure 5A). *E. coli* MG1655 and its *cpxR* null derivative demonstrated a similarly high level of sensitivity to pH 11.0 with the addition of chlorine (Figure 5). The wild-type strain harboring LHR was more tolerant (*p* < 0.05) to alkaline conditions in the presence of chlorine; however, deletion of *cpxR* or *kefB_GI_* or both diminished tolerance (Figure 5). In summary, the LHR improves bacterial survival under extreme alkaline pH only in the presence of chlorine, and the contribution of the LHR depends on the presence of *cpxR* and *kefB_GI_*.

## 4. Discussion

We hypothesized that proteins encoded by the LHR contribute to cell envelope homeostasis, thereby promoting bacterial survival under various stress conditions and reducing gene induction by the Cxp and EvgAS systems. EvgAS is a two-component regulatory system that is related to osmotic adaptation, as well as acid and antibiotic resistance, by regulating the gene encoding chaperone-like proteins and drug transporters [20,45]. The expression of the core-genome *yfdX* gene in *E. coli* is significantly induced by overexpression of the EvgA response regulator [20], and the p2 promoter of the LHR was previously reported to be induced by EvgA [14]. YfdX demonstrates chaperone-like activity preventing protein aggregation and exhibits a pH-dependent stoichiometric conversion between dimeric and tetrameric states at pH 10.0 and 7.5, respectively, which suggests its possible function involved in pH adaptation [19,46]. A deletion of *yfdX* was complemented by overexpression of the periplasmic chaperone HdeA, supporting the above prediction [47]. YfdX1_GI_ encoded by the LHR is controlled by a different promoter but also includes a predicted Sec-dependent protein secretion signal peptide at its N-terminus (SignalP, v5.0, Denmark [48]), suggesting its involvement in envelope protein homeostasis. Results of the present study contrast a previous report that expression from the LHR-promoter p2 is dependent on EvgA [14]. The p2 promoter is directly upstream of *orf1*; however, *orf1* expression levels were not dependent on EvgA (Figure 3). The role of EvgA in LHR-mediated stress resistance thus requires further investigation.

The LHR also confers significant chlorine resistance [10]. Chlorine non-selectively oxidizes cellular components, including proteins and nucleotides, and also increases the permeability of the cytoplasmic membrane [6,7,8]. Protein denaturation and aggregation in both the cytoplasm and the periplasm or membrane leakage induced by chlorine treatments can eventually lead to cell death. Combined, these observations suggest that the LHR may play a role in envelope protein homeostasis. However, our results indicate that the LHR does not mitigate the impact of the alkaline conditions on those components that activate the Cpx response (Figure 4). Expression of the LHR in *E. coli*, however, is affected by CpxR (Figure 3). The converse does not seem to be true since the presence or absence of the LHR did not impact expression of a Cpx-regulated reporter gene (Figure 2). The LHR reduces the oxidation of membrane lipids [10], and so the present study thus suggests that oxidation of membrane lipids is not an inducing signal for the Cpx pathway.

Bioinformatic analyses demonstrated that the CpxR response regulator is present in *Enterobacteriaceae*, including all of those species that include strains that possess the LHR except *Citrobacter braakii* (Figure 1). LHR-encoded proteins are among the most highly expressed proteins in *E. coli* [16]. Deletion of the CpxR response regulator did not alter LHR expression at neutral pH but increased its transcription level at pH 8.0. The Cpx pathway responds to elevated extracellular pH. Although the growth of the *cpxR* null mutant strain was comparable to the wild-type strain at neutral pH, its growth was impaired at alkaline conditions and failed to grow at pH 9.0 to 9.2 [37,40]. These results indicate the possibility that the LHR may confer tolerance to lethal conditions at alkaline pH and that the Cpx response may act as a “switch” that shifts bacterial adaptation from mild to lethal stress: Under moderate stress conditions, the activation of the Cpx pathway mitigates periplasmic stress and maintains the LHR expression at a low level to reduce fitness cost.

The adaptation to mild alkaline pH in mesophilic bacteria includes increasing acid production and ATP generation in the central carbon metabolism, enhanced expression of transporters and modification of membrane properties [21,49,50,51,52,53,54,55]. Among these, proton pumps and cation/proton antiporters are the main contributors to the ability of bacteria to maintain their internal pH. This supports growth at an external pH of 5.5–9.0 while maintaining the cytoplasmic pH in the narrow range of 7.5–7.7 [21,56,57,58,59,60,61,62]. To date, very few reports demonstrate how mesophilic bacteria cope with extreme alkaline pH. The LHR did not enhance resistance or tolerance of *E. coli* to alkaline pH; however, the presence of both the Cpx pathway and the LHR enhanced the tolerance of *E. coli* to lethal challenge with alkaline pH in the presence of chlorine. Remarkably, although expression of LHR-encoded genes was derepressed in the *cpxR* mutant of *E. coli* MG1655 during growth at alkaline pH, the LHR-mediated resistance of cultures grown at neutral pH to chlorine at extreme alkaline conditions was dependent on a functioning Cpx system. The Cpx system, in conjunction with the σ^E^ and σ^32^ responses, can sense and regulate gene expression in response to oxidative stress and high pH [26,40,63]. In *Salmonella enterica* Typhimurium, CpxR is required to cope with bacterial oxidative damage [64]. Thus, envelope stress responses to alkaline pH and oxidative stress are often linked. Because the LHR only affects survival at alkaline pH in the presence of sodium hypochlorite, one possibility is that the function of the LHR in alkaline tolerance may require oxidative stress as the inducing signal. The *E. coli* genomes encode KefB, a potassium/proton antiporter that is repressed by glutathione (GSH). Glutathione-deficient mutants of *E. coli* exhibited a reduced cytoplasmic pH at 7.35 compared to the parent strain at 7.85 [65,66]. The Kef channels are regulated by their C-terminal NAD-binding domains containing a single binding site for GSH [67]. GSH stabilizes the interdomain association between two NAD-binding folds, therefore inhibiting Kef activity [67]. GSH, a predominant low-molecular-weight thiol in most Gram-negative bacteria, has been proposed to protect cells from oxidative damage. In vitro, 1 mol GSH can react with 3.5 to 4.0 mol HOCl [68]. In *E. coli*, the glutathione-deficient strain is twice as sensitive to killing by chlorine compared to its isogenic wild-type strain [69]. The lethal level of oxidative stress can dramatically increase the amount of intracellular glutathione disulfide (GSSG) and decrease the ratio of GSH/GSSG [70]. Therefore, the addition of chlorine consumes GSH in the bacterial cell, thereby activating *kefB_GI_* and leading to cytoplasmic acidification, which protects bacteria [23]. Hence, the functional *kefB_GI_* plays an essential and specific role in protecting the bacteria from the extreme alkaline condition in the presence of chlorine.

Notably, we also observed that bacterial survival and death under extreme alkaline conditions occurred within a very narrow pH, which is opposite to its survival under extreme acidic pH. Bacterial survival at low pH is well understood as microbes encounter acid environments in many situations, including low-pH foods and during gastrointestinal transit. *E. coli* withstands exposure to the extreme acidic condition—less than 1 log cell reduction at pH 2.5 for at least 2 h—as it possesses several amino-acid-dependent extreme acid resistance mechanisms [71,72,73,74]. The acid stress response maintains the bacterial intracellular pH through the amino-acid-dependent decarboxylase/antiporter activities and protects the proteins from acid damage by inducing the periplasmic chaperone expression [73,75,76,77,78]. However, how bacteria tolerate extreme alkaline pH is poorly documented in the literature. Therefore, our study more clearly defines the range of alkaline pH that is lethal to *E. coli*: *E. coli* MG1655 withstands exposure to alkaline pH up to 11.1 with only a moderate reduction of cell counts, whereas incubation at pH 11.3 is lethal to virtually all cells (Appendix A). The mechanism behind this phenomenon remains to be investigated.

## 5. Conclusions

In conclusion, we show here that almost all *Enterobacteriaceae* known to possess the LHR also have a functional Cpx response, suggesting the possibility of the cross-talk of the Cpx pathway and the LHR. The LHR improves alkaline pH tolerance only in the presence of chlorine, and this LHR-mediated tolerance to chlorine at alkaline pH depends on the Cpx response. In contrast to the Cpx two-component regulatory system, which contributes to bacterial stress resistance under nonlethal conditions, the LHR functions to protect bacteria against lethal challenges. Our data also indicate that the CpxR response regulator negatively regulates LHR transcription under normal growth conditions to reduce the fitness cost.

## Figures and Tables

**Figure 1 microorganisms-09-00701-f001:**
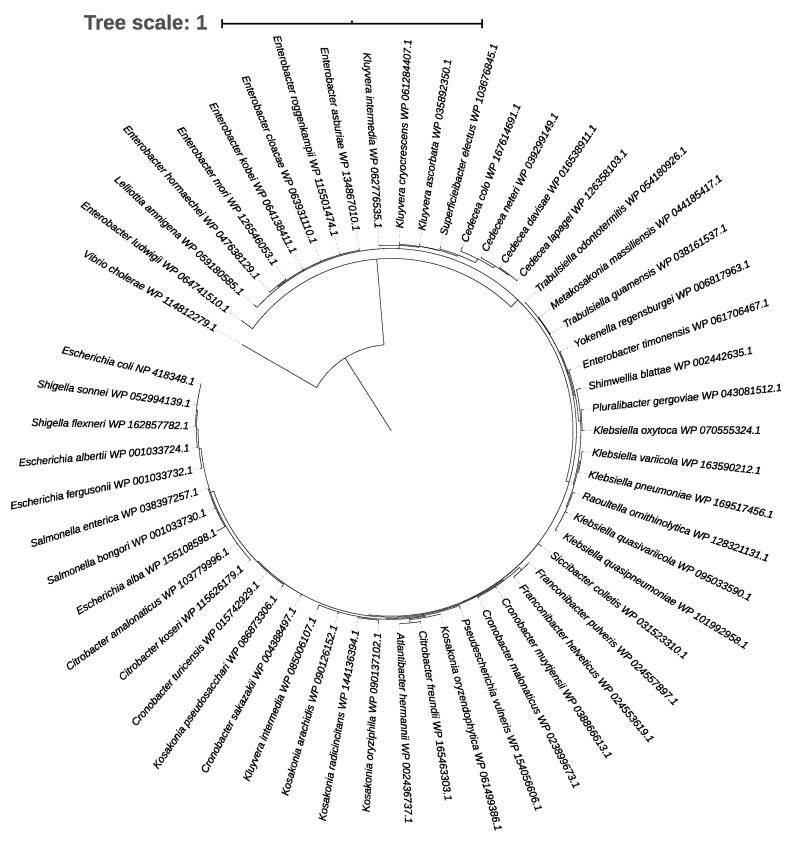
A phylogenetic tree showing the distribution of the CpxR response regulator in family *Enterobacteriaceae* with *Vibrio cholerae* as the outgroup.

**Figure 2 microorganisms-09-00701-f002:**
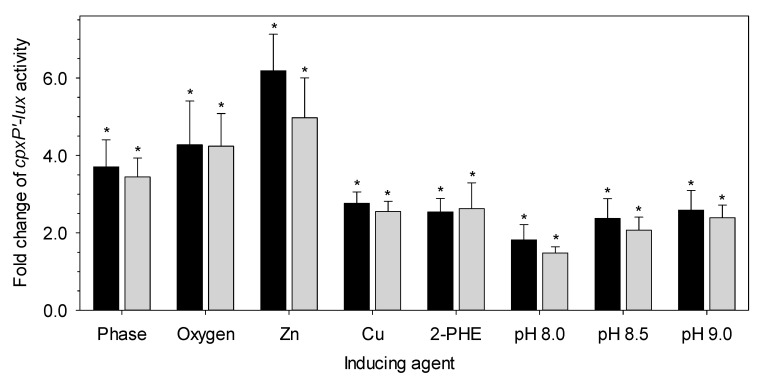
The Cpx responses in *E. coli* MG1655 (black bars) and MG1655 *lacZ*::LHR (gray bars) under different conditions. The fold change of *cpxP’-lux* activity was calculated relative to the reference conditions as follows: stationary phase vs. log phase (reference); aerobic vs. anaerobic incubation (reference); 1 mM Zn vs. no addition (reference); 1 mM Cu vs. no addition (reference); 6 mM phenylethanol (2‑PHE) vs. no addition (reference), pH 8.0, 8.5 and 9.0 vs. 7.0 (reference for all three pH values). An asterisk indicates a significant difference between the experimental conditions and the respective reference conditions (*p* < 0.05); however, *cpxP’-lux* activity was not altered (*p* > 0.05) by the presence of the locus of heat resistance (LHR). The bars represent the mean values with standard deviations as the error bars for three independent experiments.

**Figure 3 microorganisms-09-00701-f003:**
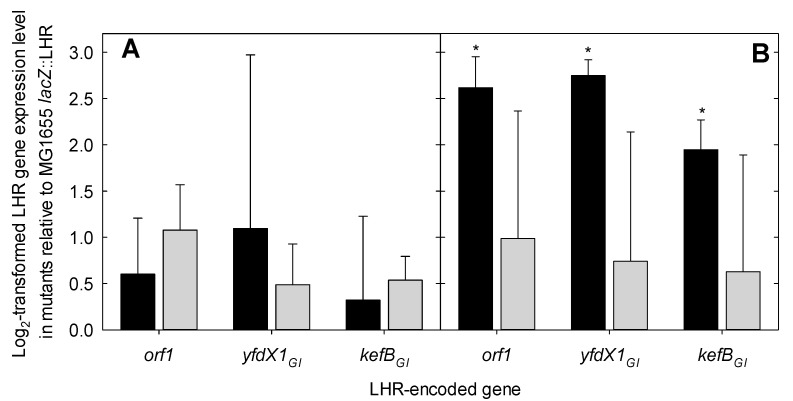
The expression level of the LHR in *E. coli* MG1655 *lacZ*::LHR with either *cpxR* (black bars) or *evgA* knocked out (gray bars) relative to the expression in MG1655 *lacZ*::LHR. (**A**) pH = 7.1; (**B**) pH = 8.0. Expression of the LHR was measured by quantification of mRNA of the genes *orf1*, *yfdX1_GI_* and *kefB_GI_* with RT-qPCR by using MG1655 *lacZ*::LHR as reference. Asterisks indicate means that were significantly different from the respective reference condition (*p* < 0.05). The bars represent the mean values with standard deviations as the error bars for three independent experiments.

**Figure 4 microorganisms-09-00701-f004:**
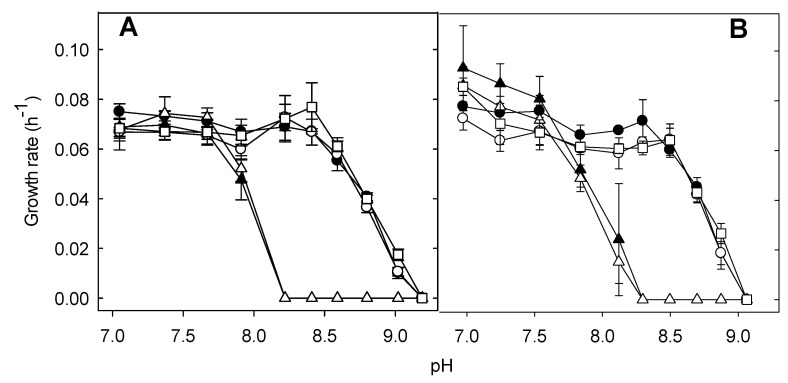
The growth rates of *E. coli* MG1655 and MG1655 *lacZ*::LHR and gene knockout derivative strains at pH values ranging from 6.9 to 9.2. (**A**) Strains were incubated in Tris-phosphate (50 mM each of Tris and phosphate)-buffered Luria–Bertani (LB) with 0.41% NaCl. (**B**) Strains were incubated in Tris-phosphate (50 mM each of Tris and phosphate)-buffered LB broth (1% NaCl). All strains were subcultured at 1:1000 from overnight cultures and incubated at 37 °C for 16 h. (●) MG1655, (○) MG1655 *lacZ*::LHR, (▲) MG1655 Δ*cpxR*, (Δ) MG1655 *lacZ*::LHR Δ*cpxR*, (□) MG1655 *lacZ*::LHR Δ*kefB_GI_*. Data are shown as mean ± standard deviation of three independent experiments.

**Figure 5 microorganisms-09-00701-f005:**
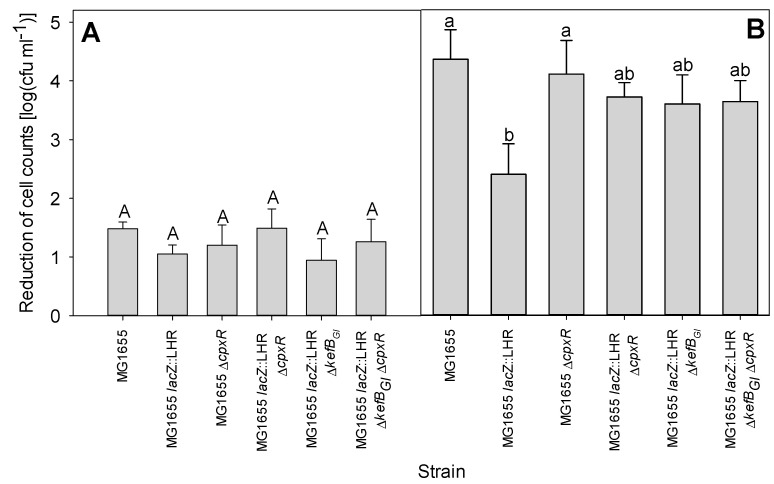
Reduction of cell counts of *E. coli* MG1655 and MG1655 *lacZ*::LHR and gene knockout derivative strains under different stress conditions. (**A**) Extreme alkaline pH killing: the overnight cultures of all the tested strains were treated with 50 mM carbonate-bicarbonate-buffered LB under pH 11.2 condition for 5 min. (**B**) Extreme alkaline pH plus chlorine killing: the overnight cultures of all the tested strains were treated at pH 11.0 with 10 mM NaClO condition for 5 min. Bars in each panel differ significantly (*p* < 0.05) if they do not share a common capital (A) or lowercase (B) letter. The bars represent the mean values with standard deviations as the error bars for three independent experiments.

**Table 1 microorganisms-09-00701-t001:** Bacterial strains and plasmids used in this study.

Strain/Plasmid	Description	Reference
*E. coli* MG1655	*E. coli* K-12 derivatives	
*E. coli* MG1655 *lacZ*::LHR	Full-length LHR with its promoter inserted into MG1655 *lacZ*	This study
*E. coli* MG1655 Δ*cpxR*::Kan	*E. coli* MG1655 with chromosomal *cpxR* replaced by the kanamycin resistance cassette	This study
*E. coli* MG1655 *lacZ*::LHR Δ*cpxR*::Kan	*E. coli* MG1655 *lacZ*::LHR with chromosomal *cpxR* replaced by the kanamycin resistance cassette	This study
*E. coli* MG1655 *lacZ*::LHR Δ*kefB_GI_*::*FRT*	*E. coli* MG1655 *lacZ*::LHR with LHR *kefB_GI_* replaced by the FRT scar site	This study
*E. coli* MG1655 *lacZ*::LHR Δ*kefB_GI_*::*FRT* Δ*cpxR*::Kan	*E. coli* MG1655 *lacZ*::LHR with LHR *kefB_GI_* replaced by the FRT scar site and chromosomal *cpxR* replaced by the kanamycin resistance cassette	This study
*E. coli* MG1655 *lacZ*::LHR Δ*evgA*::Kan	*E. coli* MG1655 with chromosomal *evgA* replaced by the kanamycin resistance cassette	This study
pJW15	Promoterless luminescence reporter plasmid containing *luxCDABE* operon, *ori_p15A_*; Kan^r^	[28]
pJW25	pJW15 plasmid containing *cpxP* promoter; Kan^r^	[28]
pLHR	Low-copy plasmid containing the LHR	[12]
pKDsg-lacZ	Plasmid containing crispr-targeting sequences for lacZ	This Study
pCas9cr4	Plasmid with *cas9* expressed under control of the P_TET_ promoter	[29]
pCP20	Plasmid enabling Flp-catalyzed excision of the antibiotic resistance gene	[30]

**Table 2 microorganisms-09-00701-t002:** Primers used in this study.

Primer	Sequence (5′-3′)	Ref. ^a)^
sgRNA-lacZ-F	GGCCAGTGAATCCGTAATCAGTTTTAGAGCTAGAAATAGCAAG	
sgRNA-lacZ-R	TGATTACGGATTCACTGGCCGTGCTCAGTATCTCTATCACTGA	
Targeting sequence	GGCCAGTGAATCCGTAATCA	
LHR-16-F	CGGTATCGCCGTCGACGACG	
lacZ-upstream	GCTGTTGCCCGTCTCACTGG	
LHR-2-R	GCCGGAATTTCCCCGTGTGC	
lacZ-downstream	GGACGACGACAGTATCGGCC	
*yfdX1*-check-F	TCGGTAAAGAAAGCGGTCAAG	
*yfdX1*-check-R	CATCGGAAGGTTGTCGGTTT	
*kefB*-P2	CATCGTGCGCTGGACGTCGACGCAAGTGGGACGCTGACCGATGGGAATTAGCCATGGTCC	
*kefB*-P1	TGGTCACGTAAGACCTGAAATGGGTTAAGGCGTGTTGATTGTGTAGGCTGGAGCTGCTTC	
*kefB*-check-F	TTGCTGGGGTATCTCTCTGT	
*kefB*-check-R	CAGCCACATCAATAGCAGGA	
*cpxR* F	CTATGCGCATCATTTGCTCC	
*cpxR* R	CATGCTGCTCAATCATCAGC	
k1	CAGTCATAGCCGAATAGCCT	[30]
*evgA* F	GACGCCTTATGTCTGTATTAC	
*evgA* R	GTTGCTGCGAATCGGTATG	
Orf1-F	GGTGATTTTCACGCTCGATG	
Orf1-R	TCGGATGACTTCTGCTGTTC	
ORF8-F	TCGGTAAAGAAAGCGGTCAAG	[34]
ORF8-R	CATCGGAAGGTTGTCGGTTT	[34]
Orf13-F	TTGCTGGGGTATCTCTCTGT	
Orf13-R	CAGCCACATCAATAGCAGGA	
*gapA*-F	GTTGACCTGACCGTTCGTCT	[35]
*gapA*-R	ACGTCATCTTCGGTGTAGCC	[35]

^a)^ Primers were designed in this study if a reference is not provided.

## Data Availability

The data presented in this study are available on request from the corresponding author.

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
