# Peer review of "Contribution of the Locus of Heat Resistance to Growth and Survival of Escherichia coli at Alkaline pH and at Alkaline pH in the Presence of Chlorine"

_microorganisms, 2021, doi:10.3390/microorganisms9040701_

Round 1

Reviewer 1 Report

Manuscript ID: microorganisms-1129633

This work looks into role of LHR into alkaline pH tolerance by E. coli and attempt to correlate this mechanism to a two-component response regulator CpxR. There is only little knowledge on bacterial alkaline pH response. Also, molecular mechanisms of LHR mediated cellular responses are largely unknown. So, this work contributes to new knowledge in the field.

Ln 4: Typo in last author name? “Michael Gänzle” instead of Michael G and Gänzle?

Abstract

Ln 18-21: Cpx seems to have role in pH >8.2-9 as well as pH 11.2 in presence of chlorine. LHR stress response seems to come into play beyond Cpx mechanism meaning only at 11.2 and when chlorine is present.

Maybe: “resistance to alkaline pH is regulated by endogenous mechanism like Cpx envelope stress response, whereas LHR confers resistance to extreme alkaline pH only in presence of additional stress like chlorine.”

Introduction

Ln 36: Maybe “15-19 kb”. After initial reports of 15 kb LHR, there are several reports of 19kb LHR, including in Salmonella and E. coli strains.

Ln 48-50: maybe, “and thus may contribute to alkaline pH tolerance, i.e., leading to survival in lethal pH range”

Ln 65: "the regulation of the genes comprising the LHR by Cpx"?

Ln 66: sentence unclear.

maybe: "whether LHR responds to alkaline pH challenge in E. coli"?

Materials and methods

Ln 71: typo. "with" repeated

Ln 72: Table 1- Center alignment of text makes it difficult to read

Ln 79, 92: Plasmids pCas9cr4 and PcP20 are not listed in Table 1.

Ln 99: Table 2- Formatting issue similar to table 1.

Maybe: Indicate "This study" in ref column for primers designed in this study?

Ln 114: why 30C incubation here, while growth and tolerance experiments were done at 37C?

Ln 117: Are culture conditions for pre-culture and the respective main culture similar? Consistency between initial sub-culture and final culture may be important, especially for gene expression experiments.

Ln 119-122: Parafilm is gas permeable. So, parafilm likely did not reduced the oxygen diffusion here.

Wouldn't shaking allow more aeration anyway? For conditions intending to reduce aeration, cultures should have been incubated without agitation.

Ln 127: typo. “aliquot”

Ln 134-135: meaning 3 biological reps? Any technical reps?

Ln 138, 168: Please provide more info on incubation conditions, shaking or not, incubation temperature. This will help correlate different findings with each other.

Ln 149-150: number of technical reps?

Ln 170-171: Why was water added?

Ln177: Sentence format.

How many technical reps?

Results

Ln 186:  No italicized format for order Enterobacterales

Ln 193: Fig 1: It is not clear if all of these isolates are LHR+

Ln 213- 214: Figure 2- Since for all conditions, Cpx expression is significantly different between reference and experimental conditions, it can be mentioned briefly in the text and omitted from the figure. * in the figure gives immediate impression of significant difference between LHR- and LHR+ strains.

Ln215: Any technical replicates?

Ln 221: Any explanation for contradiction to previous finding that LHR expression requires evgA?

Ln 226: typo “.”

Ln 233: number of technical reps?

Ln 234: “"Cpx but not LHR is necessary for growth in alkaline pH"

Ln 242: “at neutral or alkaline pH”

Ln 273, Figure 5: “Reduction in cell counts [log(cfu ml -1)]

Ln 315: typo. “periplasmic”

Ln 317: In absence of chlorine stress, growth of dLHR strains at any pH is not different than LHR+ strains. Unless the cells harvested for transcription assay and tolerance experiments are coming from same culture conditions, LHR expression data may not be able to justify the survival under extreme stress (chlorine+alkaline pH).

Ln 335: typo. “alkaline tolerance may require”. “E. coli genome encodes”

Ln 359: typo. “how bacteria tolerate”

Conclusions:

Ln 365-366:  Meaning "almost all LHR isolates also carry CpX two component system."

The term "complementary" does not seem to fit in context below pH 11.2 and absence of chlorine. So, need to clarify.

Ln 368: maybe: “.. except for in the presence of chlorine, and this response is dependent on Cpx and KefB"

Ln 369: why non-lethal conditions? dCpx isolate did not survive alkaline pH above 8.2 in non-chlorine conditions. So, pH.8.2 seems lethal to E. coli in absence of Cpx.

Supplementary data

Supplementary data

Figure S1: Since there is no significant difference between the isolates under each condition, no need for superscripts.

Reviewer 2 Report

General comments:

In this study, the full-length LHR was introduced into E. coli MG1655 strain and the role of LHR in contributing to resistance to alkaline stress was evaluated. Introducing LHR did not affect the transcription of Cpx (Fig. 2); however, CpxR repressed LHR transcription under pH 8.0 condition (Fig. 3). Introducing LHR did not improve bacterial survival to alkaline stress; also, in the cpxR-mutated background, the increase of LHR expression did not improve the bacterial survival (Fig. 4), suggesting LHR did not participate in resistance to alkaline stress. Nonetheless, results showed in Fig. 5 indicated that introducing LHR improved bacterial survival under pH 11 + 10 mM NaClO condition. Confusingly, in the absence of CpxR, the mutant strain with a high level of LHR expression did not improve bacterial survival (Fig. 5). The role of LHR in resistance to alkaline stress was not demonstrated in this study. In addition, LHR has been demonstrated to be involved in resistance to oxidizing agents including NaClO. Therefore, results from this study did not provide further information about LHR in resistance to environmental stresses.  

Other comments:

Figure 3. The experiment should be repeated, especially for yfdX1G1 and KefBG1 in cpxR mutant, and orf1, yfdX1G1, and kefBG1 in evgA mutant. 

Figure 5: The conditions including pH 8–pH 10 in the presence/absence of NaClO should be included. 

Figure 5 and supplementary result: A, a, ab showed in the Figures should be explained. 

Line 45: The previous study showed the expression of YfdX is regulated by EvgA; however, in Figure 3, the yfdX1G1 was not regulated by EvgA. What is the difference between YfdX and YfdXG1? If they are the same, why yfdX1G1 in MG1655 strain did not regulate by EvgA? Also, reference 20 should be “J Bacteriol” not “Society”.

Reviewer 3 Report

Overall, the work is very well done. I have no major comments. Hovewer, complex statistical analysis is somewhat lacking, e.g. correlation, PCA. This would enrich the work. Now, the results are presented in a unified, simple and not very interesting form for the reader.

Minor comments:

L52-66: I would distinguish more (from the new line) the purpose of the work. It fades a little here.

Table 2: next to "Ref." There is an "a)" which is not explained anywhere. What is it about?

L137-150: some references?

Figure 2: It is worth mentioning the abbreviations used on the graph when describing the conditions, i.e. Oxygen, Zn, Cu and 2-PHE. The X-axis also has no title, it is worth adding. L213: it is worth adding what statistical test was used for this analysis. And what was the n. For the Y-axis I suggest to change the unit, i.e. add more digits, e.g. every 1.5, and to harmonize the number of decimal places with other graphs (Fig. 3 has 1 decimal place in the unit).

Figure 3: The X axis does not have a title. In the caption, the A and B is enough, without the wording "panel", according to the template.

Figure 4: In the caption, the A and B is enough, without the wording "panel", according to the template.

Figure 5: X-axis has no title. L278: it is worth adding what statistical test was used for this analysis, what p was used and how many n were there.

Figure S1: X-axis has no title. In the caption, the A and B is sufficient, without the wording "panel", as per the template. It is worth adding what statistical test was used for this analysis, what p was used and how many n there were.

Reviewer 4 Report

This manuscript describes the complementary roles of the Cpx pathway and the LHR, nicely demonstrating that the LHR is needed for alkaline pH tolerance in the presence of chlorine, which also depends on the Cpx response. The authors conclude that LHR functions to protect bacteria from lethal challenges whereas the Cpx two-component regulatory system is in charge to bacterial stress adaptation under non-lethal conditions. They suggest that the Cpx system negatively regulates LHR transcription under normal growth conditions in order to decrease the fitness cost.

The manuscript is clearly written, and the experiments appear specifically conceived and well done. The discussion about the contribution of LHR to cell envelope homeostasis is of specific interest.

The authors identified in this work the extreme stress condition in which LHR expression improves bacterial survival together with Cpx system, thereby opening a new chapter towards the understanding the dynamics of stress response systems that ensure the correct cell envelope homeostasis under several stresses. They provide, in my opinion, solid data and appropriate interpretation. This nice work will be of interest to those interested in interplay among different stress response systems. My criticism is largely minor.

-Fig. 3: The authors should discuss the high standard deviations of evgA knocked out condition.

-It could be nice for the readers to have a summary figures of the hypothesized interplay between the two stress response systems.

Round 2

Reviewer 2 Report

Line 69-71: “the function of the LHR in maintenance bacterial cell envelope homeostasis” was only revealed by limited evidence in this manuscript, please consider removing it. 

As shown in the responses: LHR-mediated resistance of cultures grown at neutral pH to chlorine at extreme alkaline conditions was dependent on a functioning Cpx system. In another word, LHR only has roles in extreme conditions (pH 11.2 plus chlorine) and LHR alone did not contribute to the protection effect under this extreme condition. This information should be included in the title, the abstract section, and elsewhere in this manuscript (e.g., Line 408-410).
